# Quantitative 3D Analysis of Levator Ani Muscle Subdivisions in Nulliparous Women: MRI Feasibility Study

**DOI:** 10.3390/diagnostics14090923

**Published:** 2024-04-29

**Authors:** Nathalie Moser, Stephan Skawran, Klaus Steigmiller, Barbara Röhrnbauer, Thomas Winklehner, Cäcilia S. Reiner, Cornelia Betschart

**Affiliations:** 1Department of Gynecology, University Hospital Zurich, University of Zurich, Frauenklinikstrasse 10, 8091 Zurich, Switzerland; cornelia.betschart@usz.ch; 2Institute of Diagnostic and Interventional Radiology, University Hospital Zurich, University of Zurich, Rämistrasse 100, 8091 Zurich, Switzerland; 3Department of Nuclear Medicine, University Hospital Zurich, University of Zurich, Rämistrasse 100, 8091 Zurich, Switzerland; 4Institute of Epidemiology, Biostatistics and Prevention, University of Zurich, Hirschengraben 84, 8001 Zurich, Switzerland; 5School of Engineering, IMES Institute of Mechanical Systems, ZHAW Zurich University of Applied Sciences, Technikumstrasse 71, 8401 Winterthur, Switzerland; 6SITEM Insel-Ability, University of Berne, Freiburgstrasse 3, 3010 Berne, Switzerland

**Keywords:** levator ani, MRI, nulliparous, pelvic floor muscles, 3D-PICS, morphometry

## Abstract

Background: The levator ani muscle (LAM) is crucial for pelvic floor stability, yet its quantitative MRI assessment is only a recent focus. Our study aims to standardize the quantitative analysis of the LAM morphology within the 3D Pelvic Inclination Correction System (3D-PICS). Methods: We analyzed 35 static MR datasets from nulliparous women examining the pubovisceral (PVM), iliococcygeal (ICM), coccygeal (COC), and puborectal muscle (PRM). The PVM consists of three origin-insertion pairs, namely the puboanal (PAM), puboperineal (PPM) and pubovaginal muscle (PVaM). The analysis included a quantitative examination of the morphology of LAM, focusing on the median location (x/y/z) (x: anterior–posterior, y: superior–inferior, z: left–right) of the origin and insertion points (a), angles (b) and lengths (c) of LAM. Inter-rater reliability was calculated. Results: Interindividual variations in 3D coordinates among muscle subdivisions were shown. In all, 93% of all origin and insertion points were found within an SD of <8 mm. Angles to the xz-plane range between −15.4° (right PRM) and 40.7° (left PAM). The PRM is the largest pelvic muscle in static MRI. The ICC indicated moderate-to-good agreement between raters. Conclusions: The accurate morphometry of the LAM and its subdivisions, along with reliable inter-rater agreement, was demonstrated, enhancing the understanding of normal pelvic anatomy in young nulliparous women.

## 1. Introduction

The levator ani muscle (LAM) interacts through voluntary and involuntary movements with pelvic organs, ligaments and connective tissue and is a significant anatomical structure for pelvic floor support and function [1]. A number of ultrasound (US) and magnetic resonance imaging (MRI) studies have shown an association between LAM avulsion and the development of pelvic floor prolapse in later life [2]. Women with pelvic floor dysfunction (PFD) in general have a higher incidence of LAM avulsion and show a wider urogenital hiatus and levator hiatus, with all three factors being interrelated [3,4,5]. Further, LAM avulsions play a role in surgical failure and the recurrence of pelvic organ prolapse (POP) after various kinds of surgical prolapse repair [6,7]. 

Levator avulsions are measured semiquantitatively by scoring the LAM defects [8] that show a certain comparability between US and MR techniques [9]. More recent, 3D measurement techniques, such as estimated levator ani subtended volumes [7], volumetric segmentation [10], levator bowl volume [3], aperture size in the anterior pelvis in MRI [11] or measuring the width of the genital hiatus by US [12], have served as surrogate parameters for pelvic organ descent. However, one MR study showed that the association of levator tears and an increase in hiatus size is rather weak, in that only 12% of the variation in levator hiatus is explained by a levator defect [3]. Another study reported that the direction and angles of LAM fibers show characteristic patterns for each of the muscle’s subdivisions and that these relate to closing and lifting functions in the 2D (midsagittal) plane [13]. Clearly, more sophisticated measurements are needed before the causality of POP can be determined. 

MRI combines a multiplanar capability with pronounced soft tissue contrast and has been proven to be a very useful, objective and reproducible tool for visualizing pelvic floor anatomy and its interaction with the viscera, as well as defects of the supportive structures [14,15,16]. For a deeper understanding of abnormalities, a thorough anatomic analysis of the normal LAM in the 3D space is warranted. As a basis for comparison, the gathering of LAM reference values in young, nulliparous women is indispensable. 

Reiner et al. introduced a universally applicable 3D coordinate System, called a 3D Pelvic Inclination Correction System (3D-PICS) for MRI, which corrects for changes in pelvic inclination, producing data that can be compared intra-individually in longitudinal studies and inter-individually in cohort studies. It allows the determination of the position of any single point in the pelvis along the body axis expressed by x/y/z coordinates, taking the pubic symphysis as a fixed point (point 0/0/0) [17]. This measurement system is independent of other pelvic structures as well as individual horizontal or slant planes. Recently, the 3D characterization of pelvic ligaments [18] and pelvic organ movements in asymptomatic and symptomatic premenopausal women in dynamic MRI could be demonstrated in feasibility studies of the 3D-PICS coordinate system [19].

The aim of our study was to establish a standardized quantitative assessment of the LAM morphology in young, nulliparous women within the 3D-PICS system. To this end, 3D coordinates for the median origin and insertion points (a), angles (b) and lengths (c) of the LAM subdivisions are proposed. These parameters were determined following an anatomy-based instruction tool for each MRI plane (Table A1).

## 2. Materials and Methods

Patient selection: In the first selection process, all MR scans of the female pelvis performed during the period of 1 January 2006–31 December 2017 were screened for further evaluation. These consisted of MR scans ordered by the departments of gynecology, gastroenterology, neuroradiology, general surgery or rheumatology. The indications for the scans were, inter alia, endometriosis, diverticulosis, disc hernia and back pain. MRIs with tumors or cancer were excluded, as these conditions might distort the pelvic anatomy.

In a second selection process, the data sets were sorted according to patient age on the date of the MRI recording. We excluded all minors (<18 y/o) or patients older than 40 years of age to obtain as homogeneous a population as possible and one highly unlikely to suffer from POP. We then examined the medical records from the clinical information system to check whether the remaining MR scans met the inclusion criteria. 

Reasons for exclusion were as follows: symptoms of POP or lower urinary tract disease (LUTS), parity, history of connective tissue disease or known intrapelvic mass with effect on the positioning of the pelvic organs, such as neoplasia, fibroma or endometriosis, at the time of MRI registration. By manually reviewing the MR scans, we further excluded patients with incomplete scans or an only partially scanned pelvis, without imaging the necessary landmarks or the pelvic organs of interest (Figure 1). The images were blinded and coded as “MR#” where “#” refers to a randomly assigned number. 

MRI (Image acquisition): The static pelvic floor MRIs had been performed on a 1.5T MRI scanner (Signa^®^ EXCITE™ EchoSpeed HD or HDx. GE Healthcare, Waukesha, WI, USA; used until December 2014) or a 3T MRI scanner (MAGNETOM^®^ Skyra. Siemens Healthcare GmbH, Erlangen, Germany; used from January 2015). The reference points were measured on a T2-weighted 2D turbo-spin echo sequence acquired in the sagittal and axial plane (3T: repetition time (TR): 6034 ms, echo time (TE): 98 ms; slice thickness: 3 mm; gap: 1; flip angle: 90°; matrix sagittal: 256 × 192, transverse: 256 × 224; 1.5T: repetition time (TR): 7060 ms, echo time (TE): 86 ms; slice thickness: 3 mm; gap: 1; flip angle: 90°; matrix sagittal: 512 × 512, transverse: 512 × 512). The MRI datasets were transferred and uploaded in the OsiriX DICOM Viewer^®^ in the Institute of Diagnostic and Interventional Radiology of the University Hospital of Zurich, where the points of interest (POI) were measured (Figure A1). For further analysis of the physiologic muscle location, the scanner-based local coordinates were transformed into the pelvic 3D-coordinate System PICS. The raw data were captured and exported to Microsoft Excel. A Java^®^ tool converted these raw data to the 3D-PICS coordinate system (x′/y′/z′) inside the excel sheet. The new *x*-axis thereby included values for anterior–posterior direction, while the *y*-axis included values for the cranio–caudal and the *z*-axis for the lateral directions (left, right). In the *x*-axis, the prefix “+” stands for more posterior than point zero. In the *y*-axis, the positive prefix signifies caudally and in the *z*-axis, to the right lateral side of point 0/0/0.

The 3D-PICS System and Plane: The 3D-PICS program facilitates the derivation of 3D coordinates relative to the PICS plane, utilizing bony landmarks as references within each MR dataset. These landmarks encompassed the inferior pubic point, sacrococcygeal articulation and bilateral ischial spines. The resulting PICS plane, oriented almost horizontally along the *xz*-axis at a 90° angle to the perpendicular body axis (*y*-axis), followed the direction of gravity and prolapse in an upright posture [17]. This allowed for the calculation of angles within the pelvis and enabled the comparison of coordinates among the participants.

Identification of Anatomic Landmarks—Muscle Tracing: Detailed instruction was undertaken by the two gynecologists (NM, CB) and one radiologist (CR) to establish standardized muscle tracing in the axial, coronal and sagittal plane (Table A1). Briefly, the LAM is formed from three main muscle components, the PVM, ICM and PRM. The PVM consists of three origin-insertion pairs, namely PVaM, PPM and PAM. The COC is a triangular-shaped muscle located behind the LAM [20]. 

Each of these pelvic muscles was analyzed in DICOM-compliant image annotation software (3D-PICS DICOM Viewer, V1, April 2020). The 3D-PICS software is employed for the purpose of annotating origin and insertion points of the LAM subdivisions. 

For muscles with fan-like origins or insertions, point placement was set at 3 mm intervals and, depending on the plane, subdivided into cranial and caudal, anterior and posterior or left and right. 

Coordinates and length measurements: Origin and insertion points were expressed in 3D coordinates and the lengths of each muscle subdivision were calculated. Lengths were computed as the Euclidean distance between origins and insertions (straight line). Angles were calculated relative to the transverse, coronal and sagittal PICS planes. The directional vector of the muscles was calculated by subtracting the origin coordinates from the insertion coordinates; the normal vectors of the coronal, transverse and sagittal PICS planes were defined as the posterior, inferior and right normalized direction vector, respectively. The resultant angles, ranging from −90° to 90°, were reported. For example, muscles with a directional vector to the posterior, inferior and right were assigned a positive angle, while those pointing opposite were assigned negative angles. The PVM has one origin and specific insertion, namely the PAM, PPM and PVaM. For the lengths and angles of the PVM, a straight line was drawn between the first and last origin point. This distance was divided into 4 equal parts and the lengths or angles at the transition from part 1 to 2, part 2 to 3 and part 3 to 4 were calculated to the corresponding insertion points of the PAM, PPM and PVaM.

Statistical methods: the ratings for all 35 subjects by one rater were considered for the main analysis. 

Statistical analyses were conducted using “R 4.1.2” [21]; R Markdown [22] was used for dynamic reporting. Inter-rater reliability for all parameters was assessed using two independent raters (first and second author) on ten randomly chosen MRI datasets, by means of the intraclass correlation coefficient (ICC). In detail, ICC estimates and their 95% confidence intervals were calculated using the R package psych v2.1.9 [23] ICC(2,k), Shrout and Fleiss convention, based on a mean-rating (k = 2), absolute-agreement and two-way mixed-effects model. The limits of agreement were calculated including a 95% Wald-CI. Homoscedasticity, absence of proportional bias and normality of the differences was only assumed due to the relative low number of observations.

Summary statistics encompass means, standard deviations, medians and interquartile ranges. They are used to present coordinates, lengths and angles. Datasets from all 35 patients were used for the analyses of coordinates, lengths and angles. For the analysis of the demographics, we had incomplete data for the body weight and ethnicity; missing values were ignored for the corresponding analysis. 

The plots were created using Matlab R2019a, MathWorks, Natick, MA, USA). For the interactive graphics, the package Plotly (https://plot.ly/) [24], an online graphing platform, was used.

Regulations and clinical data: approval of the Swiss Ethics Committees on research involving humans was obtained (KEK-ZH BASEC# 2018-00102).

Magnetic resonance images were extracted from a larger project authorized by the institutional review board to conduct a study assessing pelvic floor anatomy in static pelvic magnetic resonance scans. In the present study, data were manually collected from different pre-existing MRI scans performed on young women without POP-symptoms. All data were taken from the internal digital database of the University Hospital of Zurich.

## 3. Results

### 3.1. Study Population

The mean age of the nulliparous women was 28.1 years (SD ± 6.9 years); the mean BMI was 21.9 kg/m^2^ (SD ± 3.3 kg/m^2^). Most of the women were of a European origin (*n* = 28), with five from Asia, one from North America and one of non-documented origin.

For the demographic analysis, the data for body weight (31/35 patients) and origin (34/35 patients) were incomplete.

### 3.2. Quantitative Analysis and Spatial Analysis of the 3D Muscle Points according to Origins and Insertions

In all, 93% (80/87) of all origin and insertion points were found within an SD of ≤8 mm and 32% (28/87) within an SD of ≤4 mm. Among the muscle subdivisions, the points of interest of PRM origin exhibit the most homogeneous 3D coordinates, with the lowest standard deviation (SD) in the x-(anterior–posterior) and y-(superior–inferior) axis, whereas in the *z*-axis (right–left), it is the PPM that shows the most consistent values (mean: −0.2, SD ± 1.6). It is particularly notable that the values on the *z*-axis show the greatest consistency across all muscle subdivisions compared to those on the *x*- or *y*-axis. The PAM insertion displays a marked variability in values, with an SD of ±11 mm for both the left (mean 5.1 mm) and right (mean 4.8 mm) side, as shown in Table 1. 

Spatially, the borders of the pelvic floor are formed by the PRM origin (anteriorly), the PVM (antero-laterally) and ICM origin (postero-laterally) and the COC insertion (posteriorly and cranially) and PAM insertion (caudally). 

The median muscle courses of the LAM generally slant from cranio-anteriorly towards the midline (ICM, PVM) or caudo-posteriorly (PRM, COC) (Figure 2). The most lateral points left and right are bordered by the ICM origin about 5 cm from the mid-sagittal plane (xy-plane). In the anterior–posterior axis, the ICM inserts posteriorly in the midline of the pelvis around 1 cm above the PRM in a sloping course. The PVM also shows a sloping course with a y-coordinate that is in its origin almost 2 cm lower than that of the ICM. The insertion points of the PVM are the lowest of the LAM and are found within millimeters above (PVaM) and below (PAN, PPM) the horizontal PICS plane (xz-plane). The PRM originates about 2 cm from the midline and shows a slightly ascending course in a posterior direction towards the midline. The highest point of the pelvic muscles is the insertion of the left COC, with a mean y-coordinate of 56 mm (SD: ±6.8 mm) above the horizontal PICS plane (xz-plane), while the COC origin is a few millimeters more caudal (Table 1, Figure 2).

### 3.3. Inter-rater Reliability

The ICC, which was generated for ten randomly chosen patients, shows an overall moderate-to-good agreement between the two raters (NM and SK). The highest agreement was achieved in the *y*-axis, which corresponds to the body axis and, therefore, is relevant for POP (mean ICC: 0.81, SD ± 0.13), whereas the results in the *z*-axis (left–right axis) show the least agreement (mean ICC: 0.59, SD ± 0.24). The muscle point with the lowest ICC is ICM insertion; the one with the best ICC is PPM (Table 2). However, it is important to mention that the 95% confidence interval showed considerable variability and, therefore, does not allow a definite statement about the estimated values (Table 2).

### 3.4. Limits of Agreement (LoA) 

The table (Appendix B Table A2) shows the differential bias together with the lower and upper limits of agreement (LOA) for each point of interest and their corresponding 95% Wald confidence interval. Note that measurements from different body sides were aggregated.

### 3.5. Angles

The boxplot reveals that the angles show homogeneous results with small interquartile ranges and few outliers for both body sides (Table 3, Figure 3). The most diverse results were observed for the angles of PAM and PVaM.

The posterior muscles COC and ICM generally show a mostly horizontal and lateral course (1.5° and 3.5° for the transversal plane and 60° and 55° for the sagittal plane) and might, therefore, work more in the anterior/posterior or lateral direction rather than having an elevating or depressing function.

PRM and the muscles of PVM show a slight elevation, from 14° (for PRM) up to 40° (for PAM). Their direction of action is predominantly in the anterior–posterior axis. Their angles in the sagittal plane extend from 20° to 35°. 

### 3.6. Lengths

The shortest muscle is the PVaM; the longest muscle, the PRM. Muscles on the left and right side are very similar in length (Table 4).

### 3.7. Spatial Display of the LA Muscles

A 3D visualization of the LAM is shown in Figure 2 and as an interactive graphic in a Appendix A. 

## 4. Discussion

A morphometric description of the LAM morphology using the 3D-PICS in nulliparous women is feasible and allows for the accurate identification of LAM subdivisions in the 3D space. The technique is well founded and reliable, as is reflected by the close inter-rater agreement that is important for preliminary studies and reproducibility. By providing precise 3D coordinates, our system surpasses the previous relative systems used for pelvic anatomy measurements [3,7,8,9,12,13]. Gaining a reliable spatial understanding of the normal anatomy of the female pelvic floor is crucial to understanding the development of diseases like POP. Only by defining the intact anatomy will we later be able to distinguish between a normal and abnormal anatomy, which will allow for the subsequent identification of pre-symptomatic women at risk of developing pelvic floor disorders in later life. Understanding injury mechanisms during childbirth, examining the effects of aging and studying the failure mechanisms of surgical therapies—which even today occur in one third of the patients over the long term [25]—should then become possible through precision medicine and 3D quantification.

As early as 2006, Margulies et al. [16] described specific pelvic muscle subdivisions using MRI, as each subdivision presents with a distinct morphology and characteristic features. Our research is the first morphometric study of the LAM with 3D coordinates within a young nulliparous collective and a standardized coordinate system. The point placement followed a rigorous, anatomy-based instruction and was established within an interdisciplinary team of gynecologists (NM, CB) and two radiologists (SB, CR), the latter with training in pelvic floor imaging for >15 years. After this accurate point placement instruction, moderate-to-good agreement on the detection of muscle subdivisions as well as on the usability of the 3D PICS was reached between the two younger colleagues (NM, SB) with experience of <5 years. 

A high percentage of origin and insertion points was found within a mean standard deviation of <8 mm, which is slightly higher than the mean SD of 6.1 mm reported in a recent publication on pelvic ligaments [18]. Muscle points originating in or attaching to bony structures also exhibit more homogeneous 3D coordinates. This is likely explained by the ease of identifying bony landmarks in the MRI, facilitating more precise point placement. 

The positioning of points among the study participants showed favorable agreement, especially in the *z*-axis. This can partially be explained by the fact that most of the muscles originate, insert in or are very close to the midline (*z*-axis around zero), making them less prone to anatomical variation. In contrast, the points in the x- and y-axis are mostly farther away from the 0/0/0 and, consequently, are more often subject to the anatomic variation of the pelvis, which varies according to ethnicity, age, height, body mass and gender [26]. Our sample does not allow us to draw any conclusions regarding the muscle 3D point location and length of muscles regarding ethnicity and body size as an analysis on pelvic anthropological parameters would require a much larger sample size.

The ICC showed overall moderate-to-good agreement with the best values in the *y*-axis and the lowest ones in the *z*-axis. The ICC is a relative value and thus generates lower ICC values for points showing a difference of millimeters close to point 0/0/0 than for those showing the same difference at a greater distance from point 0/0/0. Bearing that in mind, the insertion values in the *z*-axis, which are often close to the midline and, therefore, close to point zero, generate a lower ICC. Moreover, the widths of the 95% confidence interval were quite large, hindering precise statements about the values. This is likely attributed to the small number of patients included for analysis (*n* = 10). For future analyses, it would be advisable to assess the ICC in a larger patient group. 

In the context of the Limits of Agreement (LoA), it is crucial to note that any identified absolute bias, proportional bias and LoA between two raters pertain to individuals rather than distinct methods or machines. This implies that different raters may exhibit varying degrees of bias, or the same raters could reduce differences after feedback rounds. The primary objective of assessing agreement is to demonstrate that the agreement was generally good enough to build further studies and analyses on. Of course, the number of paired observations is also a relevant limitation for this method. Assumptions such as homoscedasticity, the absence of proportional bias or normality cannot be properly checked.

Although the etiology of POP is multifactorial, childbirth is known to be the single-most important risk factor [27]. Hence, it is useful to first gain data in a nulliparous collective without birth-related pelvic floor injuries, connective tissue disease or signs of aging. Even though we aimed to gather a homogeneous collective, which was possible regarding age and the absence of intrapelvic pathologies, we had some heterogeneity regarding BMI, which ranged from 17.2 to 31.2 kg/m^2^, and ethnicity. In this feasibility study, a sub-analysis on BMI and ethnicity was not possible due to the small numbers of patients, but it might be a factor to explore in further studies. The probability of prolapse in nulliparous women is very low and if it was found to occur, women were older, with a mean age of 50 years (SD: ±17–89) and heavier, with a mean body mass index of 29 kg/m^2^ (SD: ±16–64) [28]. 

A further limitation of our study is the retrospective study design with imaging not originally undertaken for morphometric research purposes. Larger patient collectives will be needed to gain normative values for the LAM muscle in healthy nulliparous women, corrected for pelvic size, gender and ethnicity [29]. Studies have reported differences in the bony architecture of the pelvis in women with and without pelvic floor disorders as well as between different ethnic groups [30,31]. Our patient group consisted largely of Caucasian Europeans.

Muscle thickness varies in both women and men, even in nulliparous women [32,33]. In our feasibility study, we concentrated on the basic, straight line morphometric parameters, such as origin and insertion point locations, and did not include other muscle parameters, such as thicknesses, curvatures or muscles shapes, which remain to be explored within the 3D-PICS measurement system.

The accuracy of point placement depends on the slice thickness and resolution of the MR images. A limitation of our study is that the MR images were acquired over a long time period from 2006 to 2017, first with a 1.5T and later on with a 3.0T MRI scanner. Technical improvements over time likely improved image quality; however, only MR images of diagnostic quality were included in the analysis. To date, no clear evidence exists that clinical MRI scans at 3.0T improve the diagnostic performance in a clinical scenario [34,35]. A dedicated high-resolution MRI scan of the pelvic floor muscles could improve the identification of the pelvic floor muscle divisions, potentially resulting in more accurate results. In combination with the mapping of other relevant structures of support, like ligaments or connective tissue, a comprehensive in vivo depiction of the pelvic support anatomy could be achieved. After establishing physiological organ and ligament position, case control studies comparing anatomic landmarks in nulliparous and parous women will give deeper insights into the pathomechanisms of POP. For the near future, we also see potential in the context of artificial intelligence. With larger datasets and machine learning, it might be possible to automatically identify specific pelvic points of interest and eliminate inter-observer bias with high precision [36]. 

## 5. Conclusions

Our study represents an innovative exploration of LAM morphometry using the 3D-PICS and provides a standardized quantitative assessment of the LAM and its subdivisions, offering insights into the spatial characteristics of the pelvic floor muscles. Our findings affirm the accuracy of MRI in detecting the LAM and its subdivisions, laying the ground for the development of a normative human pelvis atlas that also includes other supportive structures. In the future, we anticipate that such data will help us to better understand the complex anatomy of the pelvis. This new knowledge will advance the study of the physiologic ranges of supportive structures and organ locations and thus help to define threshold values so as to distinguish physiological from pathological alterations before clinically relevant prolapse is obvious. 

## Figures and Tables

**Figure 1 diagnostics-14-00923-f001:**
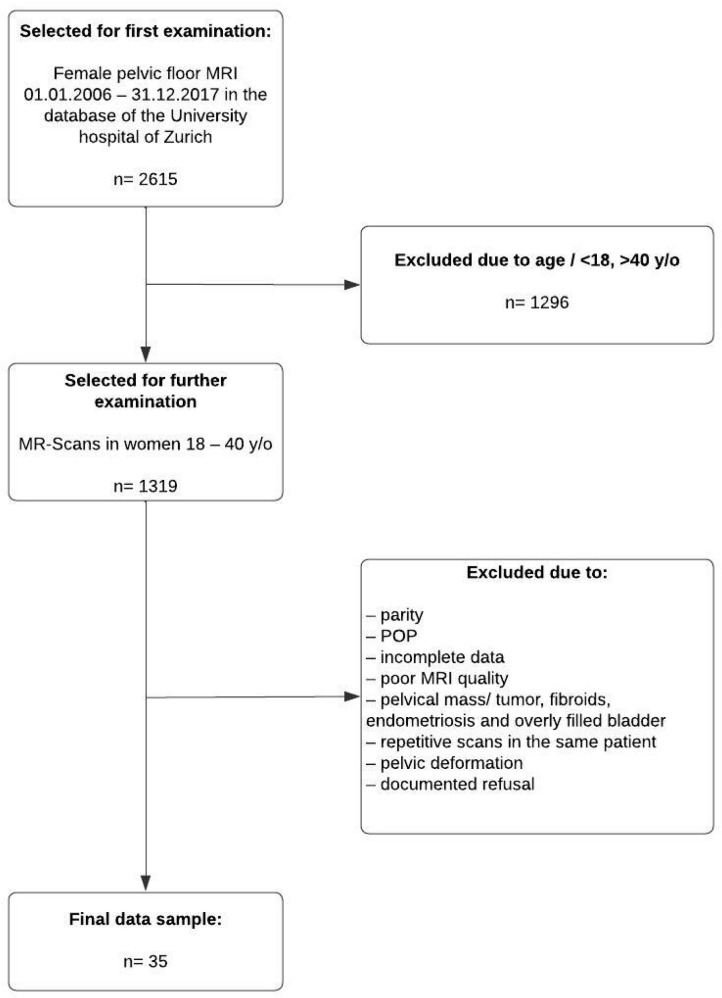
Flowchart of the patient selection process.

**Figure 2 diagnostics-14-00923-f002:**
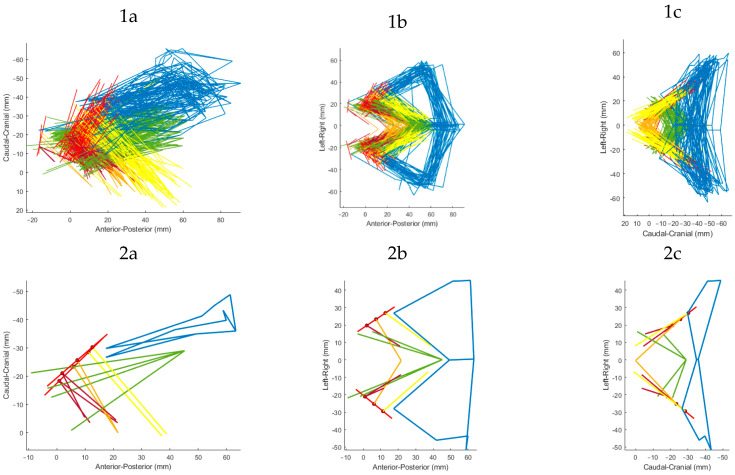
Three-dimensional visualization of the LAM subdivisions. Blue: ICM, yellow: PAM, orange: PPM, magenta: PVaM, green: PRM, bright red: PVM origin with 4 equidistances. (**1**) Muscle courses of all patients; (**2**) muscle course of one selected patient, (**a**) sagittal, (**b**) transversal, (**c**) coronal.

**Figure 3 diagnostics-14-00923-f003:**
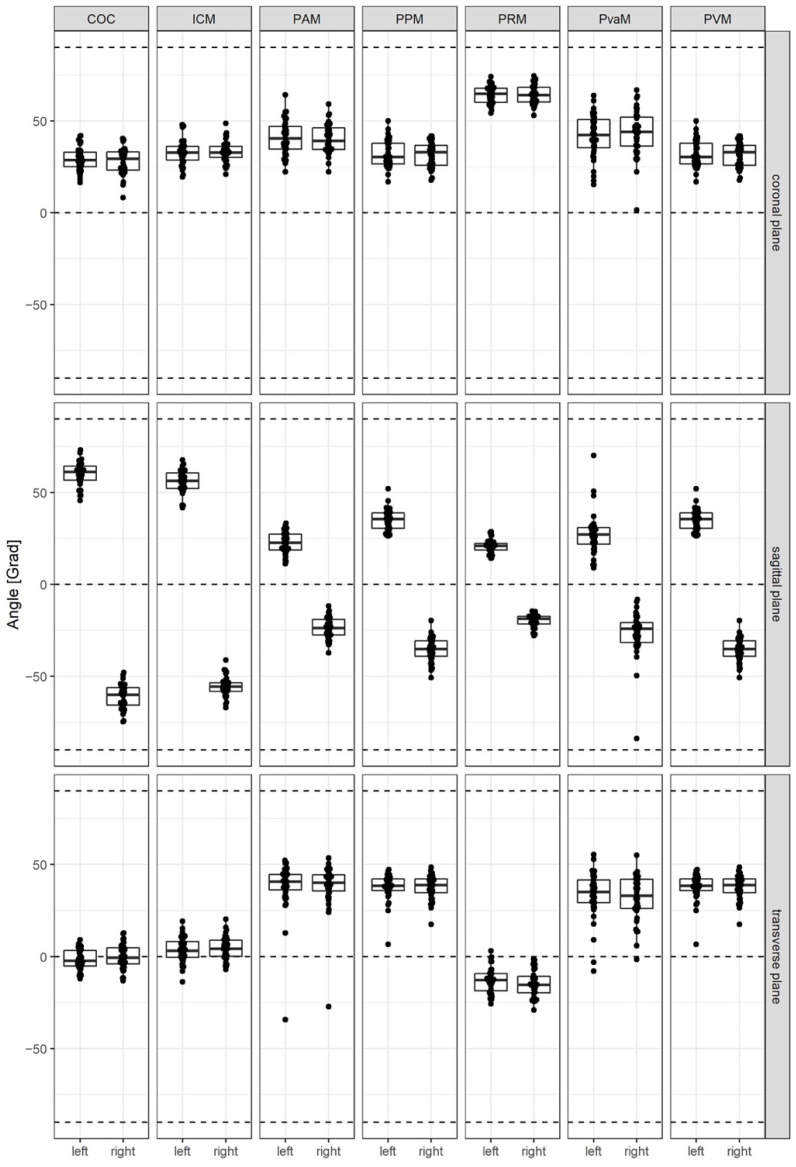
Visualization of the calculated angles in each plane.

**Table 1 diagnostics-14-00923-t001:** Numeric analysis of the marked 3D-PICS muscle points (x/y/z) in each axis. Values are expressed in mm, ±standard deviation in brackets.

Muscle	3D Coordinates Mean (±SD)																	
	Left						Midline						Right					
	1			2			1			2			1			2		
	X	Y	Z	X	Y	Z	X	Y	Z	X	Y	Z	X	Y	Z	X	Y	Z
COC origin	56.2 (6.9)	−49.5 (6.7)	−52.4 (4.3)										55.7 (5.3)	−50.6 (6.0)	51.3 (4.4)			
COC insertion	87.2 (7.9)	−56.5 (6.8)	−8.9 (3.9)	75.3 (8.3)	−45.7 (6.8)	−8.8 (3.7)							86.6 (7.2)	−56.1 (8.4)	6.4 (4.3)	73.2 (8.1)	−44.6 (6)	5.5 (5.1)
ICM origin	56.2 (6.9)	−49.5 (6.7)	−52.4 (4.3)	16.0 (7.2)	−28.7 (8.2)	−28.9 (3.9)							55.7 (5.3)	−50.6 (6.0)	51.3 (4.4)	16.3 (7.9)	−29.4 (7.9)	28.7 (4.4)
ICM insertion							73.3 (8.3)	−39.6 (6.6)	−0.2 (2.2)	52.5 (6.6)	−32.4 (7.4)	−0.1 (2.0)						
PRM origin	−7.9 (4.0)	−18.2 (3.4)	−20.5 (2.4)	4.8 (4.6)	−5.0 (3.3)	−20.1 (2.0)							−7.2 (3.3)	−16.9 (3.6)	19.1 (3.1)	5.6 (4.0)	−3.7 (2.8)	19.3 (2.5)
PRM insertion							52.5 (6.6)	−32.4 (7.4)	−0.1 (2.0)	48.9 (7.0)	−17 (7.1)	0.1 (1.9)						
PVM origin	15.2 (7.4)	−34.6 (7.6)	−32.4 (4.7)	−1.3 (5.0)	−10.5 (3.6)	−16.1 (2.9)							15.5 (7.7)	−34.5 (6.5)	32.6 (5.0)	−1.3 (4.8)	−9.7 (3.6)	14.8 (2.4)
PAM insertion	46.2 (7.9)	5.1 (11)	−7.3 (3.5)										45.8 (7.9)	4.8 (11)	6.7 (2.6)			
PPM insertion							29.0 (6.6)	3.5 (4.4)	−0.2 (1.6)									
PVaM insertion	10.3 (4.8)	−5.1 (5.2)	−13.7 (2.5)	21.2 (5.4)	−5.6 (5.0)	−9.2 (2.7)							10.2 (4.8)	−5.6 (4.9)	13.9 (2.4)	21.2 (5.3)	−5.9 (4.9)	8.7 (4.4)

**Table 2 diagnostics-14-00923-t002:** Inter-rater reliability. Data are intra-class coefficients with 95% confidence interval.

Muscle	ICC																	
	Left						Midline						Right					
	1			2			1			2			1			2		
	X	Y	Z	X	Y	Z	X	Y	Z	X	Y	Z	X	Y	Z	X	Y	Z
COC origin	0.7 (0.13–0.9)	0.83 (0.5–0.95)	0.71 (0.11–0.91)										0.43 (0–0.79)	0.94 (0.58–0.98)	0.46 (0–0.83)			
COC insertion	0.93 (0.78–0.98)	0.91 (0.72–0.97)	0.57 (0–0.86)	0.48 (0–0.82)	0.86 (0.56–0.96)	0.56 (0–0.86)							0.38 (0–0.8)	0.67 (0–0.9)	0.52 (0–0.85)	0.24 (0–0.7)	0.8 (0.41–0.94)	0.67 (0.01–0.9)
ICM origin	0.7 (0.13–0.9)	0.83 (0.5–0.95)	0.71 (0.11–0.91)	0.89 (0.6–0.97)	0.85 (0.53–0.95)	0.95 (0.84–0.99)							0.43 (0–0.79)	0.94 (0.58–0.98)	0.46 (0–0.83)	0.5 (0–0.84)	0.3 (0–0.78)	0.47 (0–0.83)
ICM insertion							0.73 (0.19–0.92)	0.87 (0.56–0.96)	0 (0–0.69)	0.9 (0.68–0.97)	0.91 (0.52–0.97)	0.22 (0–0.75)						
PRM origin	0.79 (0.37–0.93)	0.84 (0.5–0.95)	0.71 (0.13–0.91)	0.68 (0.09–0.89)	0.97 (0.9–0.99)	0.71 (0.13–0.91)							0.57 (0–0.95)	0.83 (0.48–0.94)	0.5 (0–0.82)	0.43 (0–0.81)	0.68 (0.03–0.9)	0.48 (0–0.81)
PRM insertion							0.9 (0.68–0.97)	0.91 (0.52–0.97)	0.22 (0–0.75)	0.97 (0.9–0.99)	0.88 (0.63–0.96)	0.44 (0–0.82)						
PVM origin	0.79 (0.38–0.93)	0.95 (0.73–0.99)	0.9 (0.67–0.97)	0.62 (0–0.88)	0.74 (0.22–0.92)	0.82 (0.46–0.94)							0.74 (0.23–0.92)	0.82 (0.46–0.94)	0.67 (0.08–0.89)	0.68 (0.04–0.9)	0.57 (0–0.86)	0.13 (0–0.73)
PAM insertion	0.96 (0.78–0.99)	0.81 (0.39–0.94)	0.83 (0.5–0.95)										0.97 (0.69–0.99)	0.81 (0.35–0.94)	0.47 (0–0.82)			
PPM insertion							0.94 (0.81–0.98)	0.85 (0.54–0.95)	0.82 (0.45–0.94)									
PVaM insertion	0.78 (0.36–0.93)	0.85 (0.13–0.96)	0.89 (0.67–0.97)	0.92 (0.75–0.97)	0.87 (0.15–0.96)	0.5 (0–0.84)							0.85 (0.55–0.95)	0.82 (0.12–0.95)	0.95 (0.85–0.98)	0.89 (0.56–0.97)	0.84 (0.14–0.96)	0.74 (0.23–0.91)

**Table 3 diagnostics-14-00923-t003:** Angles of the LAM in each axis, Median [1. Quartile, 3. Quartile].

Muscle	Angle °			
	Sagittal (Angle to xy-Plane)		Transversal (Angle to xz-Plane)	
	Left	Right	Left	Right
COC	61 [56.7, 64.4]	−60 [−65.8, −56.2]	−2.3 [−5, 3.3]	−0.7 [−3.9, 4.8]
ICM	56.2 [52.2, 60.7]	−55.7 [−58.3, −53.5]	3.2 [−0.5, 8.1]	4.3 [0.1, 8.9]
PRM	20.9 [18.6, 22.2]	−18.8 [−21.6, −17.4]	−12.8 [−18.6, −9.3]	−15.4 [−19.7, −10.7]
PAM	22.5 [18.8, 27.3]	−23.7 [−27.6, −19,1]	40.7 [36.2, 44.6]	40.2 [35.6, 44.5]
PPM	35.6 [30.4, 38.9]		38.5 [35.9, 42.1]	
PVaM	27.2 [21.9, 30.9]	−24.3 [−31.7, −20.7]	35.2 [29.3, 41.6]	33 [26.1, 42.1]

**Table 4 diagnostics-14-00923-t004:** Mean lengths of the muscles in mm, ±standard deviation in brackets.

Muscle	Left	Right
COC	50.8 (7.2)	52.2 (7.1)
ICM	49.3 (4.7)	49 (5.3)
PRM	58.1 (5.7)	57.3 (6.1)
PVM (average)	42.3 (6)	41.8 (5.3)
PAM	54.8 (8.6)	54.1 (7.4)
PPM	42.3 (6)	41.8 (5.3)
PVaM	19.9 (5.0)	18.8 (4.7)

## Data Availability

Following the open access policy of the University of Zurich and the Swiss National Research Foundation, data will be available after publication via open access on the University of Zurich Open Access Repository and Archive (ZORA), https://www.zora.uzh.ch/.

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
