# Peer review of "Quantitative 3D Analysis of Levator Ani Muscle Subdivisions in Nulliparous Women: MRI Feasibility Study"

_diagnostics, 2024, doi:10.3390/diagnostics14090923_

Round 1
Reviewer 1 Report
Comments and Suggestions for Authors
The study is interesting. However, an important bias is represented by having used different equipment which may not make the results comparable. I believe that it is necessary for the Authors to clarify and discuss this bias also using references to the technical data sheets of the equipment.
Author Response
We have great appreciation for the reviewer and his comments on our manuscript and would like to thank him. In order to address his concern we would like to describe our equipment properly. In case we did misunderstand the question, we apologize and are of course willing to reanalyze the request.
|
Usage of different euqipment |
For the image acquisition all MRIs have been perfromed on either a 1.5T MRI scanner (Signa® EXCITE EchoSpeed HD or HDx. GE Healthcare, Waukesha, Wisconsin, USA; used until 12/2014) or a 3T MRI scanner (MAGNETOM Skyra. Siemens Healthcare GmbH, Erlangen, Germany; used from 01/2015). The MRI scans were screened and manually reviewed. MR-scans which did not cover the necessary landmarks or where the landmarks were not visible due to degraded image quality were excluded. The MRIs from both scanners were comparable. Although MR images at 3.0T typically have a higher in-plane resolution compared to 1.5T images, up to date there is no clear evidence that clinical MRI scans at 3.0T improve the diagnostic performance in a clinical scenario (References: 37, 38). Also our MRI scans were performed for clinical purposes and with the same slice thickness (3 mm) at 1.5 T and 3.0T, which is a crucial parameter for precisely delineating structures (as the levator ani) that run through the acquisition plane. The detailed imaging parameters are listed in the Material and Methods section. Nevertheless, the use of two different MRI field strength may have introduced a certain bias. We included a respective statement in the limitations section (highlighted in green).
|
We hope that we have adequately addressed your comments and that the changes made to the manuscript (highlighted in yellow and green) also positively influence your concerns which you expressed in the general evaluation.
Sincerely,
Nathalie Moser, Stephan Skawran, Klaus Steigmiller, Barbara Röhrnbauer, Thomas Winklehner, Cäcilia Reiner, Cornelia Betschart
Reviewer 2 Report
Comments and Suggestions for Authors
Manuscript title: "Quantitative 3D Analysis of Levator ani Muscle Subdivisions in Nulliparous Women: MRI Feasibility Study"
The authors provide valuable insights of normal levator ani muscle anatomy and subdivisions using a previously outlined magnetic resonance imaging-based three-dimensional assessment system. The study is original, well-structured, and relevant to the field of gynecology. Its main strengths are detailed rationale and methodology, allowing for a high degree of confidence in result reproducibility. The areas of improvement include sample size (i.e., collecting and analyzing data after 2017), limits of agreement reporting (i.e., Bland-Altman plot), time required for the assessment and potential clinical implementation. The figures and tables are appropriate. The conclusions are consistent with the evidence provided, given the aforementioned areas of improvement. The ethics statements and data availability statements are adequate.
Author Response
We would like to sincerely thank the reviewer for reviewing our manuscript and for the constructive comments provided. We have thoroughly examined your feedback and would like to address it in detail in the following sections.
In case we did misunderstand any of the questions, we apologize and are of course willing to reanalyze the request.
|
Sample size |
This is a technique development study to improve the understandig of Leavtor ani muscle in MRI and it is a first quantitative description of the muscles morphometry within the 3D PICS system. We acknowledge the relatively small cohort. |
|
Limits of agreement reporting |
A Bland-Altman plot was examined by the authors on all aggregated data points to assess overall absolute bias, linearity of bias, and proportional bias (partially investigating a potential Simpson's paradox), but was not included in the manuscript.
à We have included a Table in the Appendix section (Table A2: absolute (mean) bias between two rates, the lower and upper LoA with the 95% Wald Confidence Interval in brackets) Additionally, we have incorporated comments into the material & methods, results, and discussion sections. We kindly ask you to review the highlighted paragraphs (yellow) for any necessary revisions.
|
|
Time required for the assessment and potential clinical implementation |
At the outset of the project, the primary author familiarized herself with the anatomy of the levator ani muscle. The anatomical landmarks were positioned based on anatomy-based instruction (Table A1). Initially, the marking of these points took slightly longer; however, a steep learning curve was observed over time, allowing for the efficient identification and marking of the respective points in a short period. Marking all points of interest (n=29) thus took less than 30 minutes. Subsequently, the marked points, identified in the Osirix DICOM Viewer, were converted into the 3DPICS format.The raw data was captured and exported to Microsoft Excel. A JavaÒ tool converted this raw data to the 3D-PICS coordinate system (x’/y’/z’) inside the excel sheet. As our project evolved, the 3D-PICS program now allows the direct extraction of the 3D-coordinates relative to the PICS plane. Bony landmarks have to be set once for each MR dataset as a reference frame. This additionally simplified the process. At present the Identification of anatomic landmarks is not yet suitable for clinical implementation. We set groundwork for further research and future studies with lager sample size. The next research project undertaken by part of the the study group involves automating segmentation and creating a user-friendly application for 3D-PICS. This application aims to facilitate widespread and straightforward utilization of the 3D-PICS technology. A preoperative staging using MRI, which provides an accurate depiction of pelvic anatomy, holds promise for enhancing surgical precision and improving patient outcomes. Moreover this new knowledge will advance the study of the physiologic ranges of supportive structures and organ locations and thus help to define threshold values so as to distinguish physiological from pathological alterations before clinically relevant prolapse is obvious. |
We hope that we have adequately addressed your comments and that the changes made to the manuscript positively influence your concerns which you expressed in the general evaluation.
Sincerely,
Nathalie Moser, Stephan Skawran, Klaus Steigmiller, Barbara Röhrnbauer, Thomas Winklehner, Cäcilia Reiner, Cornelia Betschart
Round 2
Reviewer 1 Report
Comments and Suggestions for Authors
The authors have conducted a sufficient review
Reviewer 2 Report
Comments and Suggestions for Authors
The authors have provided sufficient responses to the reviewer's queries. The manuscript's quality has been further improved.